# Integrated Oxygen Consumption Rate, Energy Metabolism, and Transcriptome Analysis Reveal the Heat Sensitivity of Wild Amur Grayling (*Thymallus grubii*) Under Acute Warming

**DOI:** 10.3390/biology14121718

**Published:** 2025-12-01

**Authors:** Cunhua Zhai, Ziyang Wang, Luye Bai, Bo Ma

**Affiliations:** 1College of Life Science and Technology, Harbin Normal University, Harbin 150025, China; 2Heilongjiang River Fishery Research Institute, Chinese Academy of Fishery Sciences, Harbin 150070, China; 3College of Fisheries and Life Science, Dalian Ocean University, Dalian 116023, China; 4Research Station for Wild Scientific Observation on Fishery Resources and Ecological Environment Protection, Jiamusi, Ministry of Agriculture and Rural Affairs, Jiamusi 154000, China; 5Scientific Observation Station of Fisheries Resource and Environment in Heilongjiang River Basin, Ministry of Agriculture, Harbin 150070, China

**Keywords:** *Thymallus grubii*, warming, oxygen consumption rate, transcriptome

## Abstract

Global warming is heating up cold mountain rivers. We wanted to know how much heat the Amur grayling, a cold-water fish, can take before its body starts to fail. Fish were placed in tanks with the same water speed but six temperatures from 9 °C to 24 °C. We tracked their breathing rate, hemoglobin, energy stores, and liver activity. Breathing quickened up to 15 °C then dropped, showing the fish were stressed. Hemoglobin rose to 18 °C but fell sharply above that. Sugar and starch stored in the liver first increased then plummeted, while lactic acid (a fatigue signal) built up. At 15 °C the fish still used sugar and protein for fuel, but at 21 °C they had to switch to fat breakdown, a sign of emergency metabolism. These changes point to a critical comfort zone between 12 °C and 15 °C. Keeping river temperatures below this range through shade, flow protection, and reduced heat inputs will help wild Amur grayling survive hotter summers.

## 1. Introduction

Acute warming is one of the most significant environmental challenges that wild fish may encounter [1]. Because fish are poikilothermic, water temperature governs their physiology and life-cycle events, making them dependent on natural cycles of flow and temperature for successful reproduction and survival [2]. Currently, water temperature effects on cold-water fish constitute a major research area due to the catastrophic mortality that occurs when temperatures exceed their thermal tolerance [3]. Evidence has confirmed that temperature change affects the survival, reproduction, growth performance, and physiological parameters of cold-water fish. For example, the existence, nutrition, and development of Atlantic salmon (*Salmo salar*) are diminished after exposure to high water temperatures (>25 °C) [4], while exposure to high temperature stress (>21 °C) evokes immune responses and histopathological damage in rainbow trout (*Oncorhynchus mykiss*) [5]. The adverse impacts on the antioxidant systems and the histopathology were observed when tsinling lenok trout (*Brachymystax lenok tsinlingensis*) were exposed to heat stress (>24 °C) [6]. While water temperature plays a crucial role in the survival and distribution of stenothermal species, there is a scarcity of research on the stress response to temperature in the grayling species.

When fish suffer from acute temperature changes, fish species adjust a series of physiological reactions to build an adaptation to current survival conditions [7]. Among them, the most direct response is energy metabolism mode alteration. Metabolic rate, measured as oxygen consumption rate (MO_2_), is widely used to determine physiological responses to an energy-demanding process [8]. Brett [9] studied the swimming capacity and oxygen consumption rate of sockeye salmon at five different temperatures and different flow velocities, and the results showed that the activity and metabolism of sockeye salmon were obviously limited by oxygen supply above 15 °C. In addition, the MO_2_ of bat rays (*Myliobatis californica*) was particularly temperature-sensitive between 14 °C and 20 °C [10]. Therefore, exploring the relationship between respiratory rate and various environmental factors will help us to further understand the physiological and metabolic patterns and energy requirements of fish.

In poikilothermic animals, the control mechanisms of metabolism processes are attributed to enzymatic regulation [11]. Energy metabolism is considered the most important factor that helps adaptation to changing environmental conditions. Under stress conditions, fish species affect the glycometabolism process by regulating the enzyme’s activities [12]. The glycolysis pathway provides energy essential for life activity [13]. Under the aerobic condition, pyruvate enters into the mitochondria and then undergoes the oxidative phosphorylation process. Glycolysis starts with hexokinase (HK) phosphorylating glucose to glucose-6-phosphate (G6P). Subsequently, pyruvate kinase (PK) phosphorylates ADP into ATP to meet the requirement for energy supplement. Succinate dehydrogenase (SDH) couples oxidative phosphorylation with electron transport to involve the process of aerobic metabolism, thereby producing sufficient energy for physiological responses [14]. In vertebrates, the reduction of pyruvate to lactate is catalyzed by lactate dehydrogenase (LDH), an indicator of anaerobic glycolytic capacity [15]. In addition, glucose, lactate, and glycogen contents could also reflect the stress conditions [16]. Therefore, studying energy-related metabolite contents has always been a direct response for the physiological processes of fish.

In natural environments, the preferred temperatures of grayling vary seasonally across different life-history stages, requiring specific thermal windows to optimize growth, reproduction, and the survival of eggs and larvae [17,18]. During summer, the optimal temperature range for European grayling is 12.0–15.7 °C [19], whereas in winter it survives under ice at temperatures close to 0 °C [20]. Spawning occurs in early spring at low temperatures, with successful reproduction requiring 7.8–11.5 °C [21]. The physiological requirements of grayling are tightly coupled to temperature throughout their life history, exhibiting highly specific physiological thermal limits that are lower than those of other cold-water salmonids such as brown trout (*Salmo trutta*), Atlantic salmon (*Salmo salar*), and brook charr (*Salvelinus fontinalis*) [22]. This elevated heat sensitivity indicates that prolonged exposure to temperatures exceeding their physiological tolerance could be detrimental.

Amur grayling (*Thymallus grubii*) is endemic to the Amur River system of China and Russia. Within China, it occurs exclusively in the upper mountain reaches of the Amur, Nenjiang, and Songhua rivers and is a cold-water freshwater fish species with high nutrition value. Wild resources live in mountain streams, accompanied by high-velocity waters and lower water temperatures. The adult grayling prefers water velocities between 20 and 45 cm/s [23]. The temperature range allowing survival of the grayling species is extremely narrow (8–18 °C) [18]. It has been listed in the “China Red Data Book of Endangered Animals” [24] due to Amur grayling being easily affected by external environmental changes and human activities [25]. Now, field surveys at summertime (August) have documented that Amur grayling actively abandoned habitats where water temperatures exceeded 20 °C and relocated to adjacent cooler refuges whenever these were accessible. Despite the concerning trends in water temperatures, there is a paucity of knowledge regarding the physiological response of Amur grayling. Therefore, the factor of most concern affecting the distribution and survival of grayling in the future is the increases in water temperatures where they reside [26,27]. In order to more accurately reflect the effect of temperature change for the fish, the combined effect of water temperature and flow velocity should be considered to implement a more realistic simulation of its environment [28]. Therefore, we have taken the Amur grayling as an example to investigate the metabolism response mode to warming in grayling while monitoring velocities.

Based on the above, the objectives of the present study were to (1) explore the effect of fixed flow velocity (the suitable flow velocity for *Thymallus grubii*) on metabolite levels and oxygen consumption rate (MO_2_) of *Thymallus grubii* under different water temperature scenarios (9, 12, 15, 18, 21, and 24 °C) and (2) investigate the possible mechanism of MO_2_ change in adult *Thymallus grubii* under key high-temperature points (9, 15, and 21 °C) according to the results of MO_2_, providing initial insights into the physiological regulatory mechanisms employed by *Thymallus grubii* to cope with warming. Our study will clarify the adaptation feature to acute warming in *Thymallus grubii* and pave a path for future advancements in the habitat restoration of cold-water fish.

## 2. Material and Methods

### 2.1. Experimental Animal Preparation and Breeding

Animal care laws and guidelines (Directive 2010(63)EU) were followed for all treatment procedures and were submitted by the Laboratory Animal Ethics Committee of the Research Institute of Fisheries of the Heilongjiang River (No. 20231028-001). We confirm that the study is reported in accordance with ARRIVE guidelines.

All experimental fish were collected by our team using angling and cast nets in the Huma river from the upper Amur River (Figure 1a) in China in autumn and then transported to the indoor culture facility (Harbin, China). All fish were habituated for 3 d prior to the formal experiment in a temperature-controlled recirculating water tank (80.5 cm × 48 cm × 39 cm) to remove transport stress and maintain wild habits [29]. Fish were not fed during this period. The tap water was aired for 5 h and changed on a daily basis, the temperature of the water was maintained at about 9 °C, and the pH value was in the range of 6.8 ± 0.12. In addition, dissolved oxygen (DO) was controlled at 8.2 ± 0.3 mg/L and the light cycle was set to 12 h light/12 h dark. In this study, the water quality was analyzed using a Water Quality Meter (Thermo, 4-Star, Waltham, MA, USA).

### 2.2. Experimental Facility Overview

Fish were tested in a 10 L (40 cm × 10 cm × 10 cm) rectangular swim chamber of the Loligo system (Loligo system SY28060, Viborg, Denmark) recirculating swim channel. An automatic water temperature controller was used to control the water temperature during the experiment. A multiparameter probe (HACH HQ30d, Loveland, CO, USA) was used to monitor DO. Subsequently, the data were transmitted to the Witroxview program (Loligo System, Viborg, Denmark). The relationship between flow velocity in the experimental tank measured by tachometer and frequency adjusted by the controller was shown in Appendix A. 

### 2.3. Experiment Designs

Sixty healthy fish were chosen and indiscriminately allocated to six tanks (ten fish per tank) for the MO_2_ measurement assay for Amur grayling with similar body lengths (16.98 ± 0.34 cm). Zhai et al. [30] have shown that the optimal flow velocity for Amur grayling with body lengths of 16.98 ± 0.34 cm is determined to be 34 cm/s. Because 9 °C is the ideal temperature for sustaining the healthy metabolic activity of Amur grayling in autumn, 9 °C was used as the initial temperature of Amur grayling. Subsequently, the water temperature in one of the tanks remained unchanged at 9 °C, which was defined as the control group, and the water in the others were increased in temperature at a rate of 2 °C/1 h until the target temperature (12, 15, 18, 21, and 24 °C) was reached (Figure 1b). Eight fish were randomly selected from each temperature group after 5 min and were transferred to a 10 L rectangular swim chamber (Loligo system) containing the target temperature (9, 12, 15, 18, 21, and 24 °C) maintained by the automatic water temperature controller (Figure 1c). Then, eight fish were kept at target temperature for 40 min with the same flow velocity (34 cm/s) until oxygen consumption data were collected. Oxygen concentration was maintained above 90% saturation (>8.6 mg O_2_ L^−1^) throughout the acclimation and experimentation period. Temperature and oxygen saturation were measured continuously (1 s response time) using calibrated oxygen and temperature electrodes. The fish had enough time to adjust to the tank’s new environment to avoid stress, which was controlled remotely by a computer with cameras and free of human interference. Six fish were chosen randomly from each temperature group and immediately anesthetized using 0.02% Methane Sulfonate-222 (MS-222; Sigma, St. Louis, MO, USA). After adequate anesthesia, fish were weighed immediately (Table 1), while the blood, liver, and tail muscle tissues were immediately collected. In the following studies, the serum, liver, and tail muscle tissues of each group were obtained to determine blood glucose, glycogen, lactic acid, and creatine phosphate contents. Under the ice bath, extraction solution was added to the second portion of liver tissue to homogenate for SDH, PK, HK, LDH activities, and liver glycogen determinations. The third portion of liver tissue was placed in liquid nitrogen and then stored at −80 °C for quantitative real-time PCR and transcriptome analyses.

MO_2_ was calculated using the equationMO_2_ = V × (d(DO)/dt)/M (1)
where V is the volume of the respirometer (L), M represents the mass value of the subjected fish, and d(DO)/dt means the regression equation slope (mg O_2_/min). DO was recorded every 1 s.

At the end of each 40 min measurement period, the chamber was flushed with fish-free and temperature-matched water, and the dissolved oxygen was recorded for 5 min. This background MO_2_ value (always <1% of fish MO_2_) was subtracted from the preceding fish recording.

The thermal coefficient (Q_10_), which represents the sensitivity of an organism to temperature variations, was estimated for Amur grayling at different temperatures. Q_10_ was calculated asQ_10_ = (K2/K1)^10/(T2−T1)^
where K1 and K2 are the oxygen consumption at temperatures T1 and T2, respectively.

### 2.4. Determination of the Enzyme Activity of Glucose Metabolism, the Content of Substances of Energy Metabolism, and Hemoglobin Concentration Analysis

Hepatic SDH, PK, HK, LDH activities, and the content of glycogen, lactic acid, and creatine phosphate were measured utilizing kits purchased from Shanghai Jianglai Industrial. Blood glucose was measured using chemistry analyzers (AU5800, Beckman Coulter, Brea, CA, USA). Muscle and liver samples were homogenized and fixed with cold extraction solution at a ratio of 1:9 (*w*:*v*). After it was centrifuged at 14,000 rpm for 10 min at 4 °C, the fresh homogenate was collected into sterile tubes. The absorbance microplate reader (SpectraMax Plus 384, Molecular Devices, San Jose, CA, USA) was used to determine optical density values at 600 nm (SDH), 340 nm (PK), 340 nm (HK), 450 nm (LDH), 450 nm (lactic acid), 510 nm (glycogen), and 520 nm (creatine phosphate), respectively. Hemoglobin concentrations (HGB g/L) were determined through measurement by the Auto Hematology Analyzer (BC-2800Vet, Mindray, Shenzhen, China).

### 2.5. Project for the Construction of cDNA Libraries and Transcriptome Sequencing

The 80 mg fresh liver tissue samples were collected using sterile scissors and forceps. Total RNA samples were extracted using standard TRIzol reagent (Invitrogen, Waltham, MA, USA) according to the manufacturer’s protocol. The NanoDrop ND-1000 spectrophotometer (Thermo Fisher Scientific, Wilmington, DE, USA) was used to evaluate the quality (purity, concentration, and integrity) of the RNA. RNA integrity was detected using 1% agarose gel electrophoresis. The qualified samples were used to build sequencing libraries. mRNA was enriched using oligo (dT) beads and digested into short segments using buffer. Then, DNA polymerase I, RNase H, dNTP, and buffer were used to synthesize first-strand cDNA. Following purification and end repair, cDNA fragments were sequenced based on the Illumina HiSeq^TM^2500 platform (Allwegene Tech Co., Ltd., Beijing, China).

### 2.6. Quality Control, De Novo Assembly, and Annotation

Raw data were filtered by removing the reads of adapters, containing > 10% unknown bases (N), and low-quality reads containing > 50% low-quality (Q-value ≤ 20) bases. The clean data were de novo assembled using Trinity v2.4 software. The assembled unigenes were aligned by using the Blastx program of the BLAST package (v.2.15.0) to obtain comprehensive gene function information based on a public database, including Nr (NCBI non-redundant protein sequences) and an evolutionary genealogy of genes: Non-supervised Orthologous Groups (eggNOGs), Protein family (Pfam), Swiss-Prot (a manually annotated and reviewed protein sequence database), and Kyoto Encyclopaedia of Genes and Genomes (KEGG) and Gene Ontology (GO).

### 2.7. Differentially Expressed Gene (DEG) Identification

Raw reads for each sample were assembled with reference genomes using StringTie (v3.0.2) and then fragment per kilobase of transcript per million (FPKM) values were calculated to analyze the relative gene expression amount. To compare the DEGs among different temperature treatments, six samples from 9 °C, 15 °C, and 21 °C conditions were used for transcriptome analysis. Clean data were analyzed using DESeq2 to identify the DEGs between the 9 °C group and 15 °C group and between the 9 °C group and 21 °C group. Q values < 0.005 and |log2(fold change)| ≥ 1 were regarded as the threshold for DEG filtration.

### 2.8. Quantitative Real-Time PCR (qRT-PCR) Verification

To validate the accuracy of sequencing data, typical DEGs in comparison (9 °C vs. 15 °C and 9 °C vs. 21 °C) were selected for qRT-PCR verification (Appendix A). Primer pairs were designed based on Primer premier 6.0 software and synthesized by Sangon Co., ltd., Shanghai, China. The β-actin gene was selected as the internal reference gene. The first strand cDNA was synthesized using Prime Script^TM^ RT reagent kit with DNA Eraser (TaKaRa, Shiga, Japan). The reaction mixture contained 5 μL 2 × TB Green Premix Ex Taq II (Tli RNaseH Plus), 0.4 μL of each forward and reverse primer, 3 μL sterile distilled H_2_O (dH_2_O), 1 μL template cDNA, and 0.2 μL 50 × ROX Reference Dye II (Roche, Basel, Switzerland). qRT-qPCR was performed in the ABI 7500 real-time PCR instrument (Thermo Fisher Scientific). Detailed cycle parameters were set at 95 °C for 180 s, 40 cycles of 95 °C for 5 s, 60 °C for 15 s, and 72 °C for 30 s. The 2^−ΔΔCT^ method was used to calculate relative gene expression levels.

### 2.9. Statistical Analysis

The statistic processing procedure was implemented based on IBM SPSS Statistics 23.0 software and experiment data was shown as the mean ± standard deviation (SD). The normality and variance homogeneity of the data were assessed through the Kolmogorov–Smirnov and Levene tests, respectively. Subsequently, one-way ANOVA was utilized to determine the significance of differences. *p* < 0.05 was regarded as the significance threshold. Plots were visualized using GraphPad Software (Version 8.4.3, USA). The unary linear regression equation model was utilized to analyze the relationship between water temperature and MO_2_. For the visualization process of transcriptome analysis results, standard bioinformatics script was run in the R package (R 4.4.2) and the process obtained technological support from the professional cloud platform provided by the third-party company (Allwegene Tech Co., Ltd., Beijing, China).

## 3. Results

### 3.1. Oxygen Consumption Rates at Different Temperatures

MO_2_ was calculated using equation (1). MO_2_ increased initially and then decreased with increasing water temperature (Figure 2). There was a statistical difference between the 12 °C group and 15 °C group, and between the 18 °C group and 21 °C group (*p* < 0.05). Q_10_ was highest between 12 and 15 °C (Q_10_ = 5.30) (Table 2). MO_2_ increased significantly at 12-15 °C (Q_10_ = 5.30), then decreased at 18-21 °C (Q_10_ = 0.05). Environmental temperatures of 15 °C and 21 °C were an important node to explore the pattern of change in the respiratory metabolism of *Thymallus grubii* at high temperatures. This provided a basis for selecting 15 °C and 21 °C as the test group of transcriptome analysis.

### 3.2. Effects of Temperature Changes on Physiological Indexes of Amur Grayling

LDH activity significantly increased in the liver of the Amur grayling at 9-15 °C, then decreased with the increase in water temperature (*p* < 0.05) (Figure 3a). HK activity gradually increased then decreased with the increase in temperature, but there was no obvious difference observed between the 9 and 12 °C group and between the 18 and 24 °C group (*p* > 0.05) (Figure 3b). Conversely, SDH activity gradually decreased with the increase in temperature, but no significant difference was observed between 9 °C and 12 °C (*p* > 0.05) (Figure 3c). PK activity was significantly elevated (*p* < 0.05) from 9 to 15 °C and then markedly decreased (*p <* 0.05) under the 18–24 °C condition (Figure 3d).

Hepatic glycogen content rose markedly at 9–12 °C but decreased markedly at 15–24 °C (*p* < 0.05). Similarly, the glycogen contents of muscle rose significantly at 9–15 °C but decreased considerably in the 18-24 °C group (*p* < 0.05) (Figure 3e,f). With increasing temperature, muscle creatine phosphate was not significant at 9–18 °C (*p* > 0.05) but then significantly decreased at 21–24 °C (Figure 3g). Blood glucose levels declined gradually with the increase in experimental temperature, but the difference was not significantly obvious between 15 and 18 °C and between 21 and 24 °C (*p* > 0.05) (Figure 3h). Conversely, plasma and muscle lactate levels rose as temperature increased (*p* < 0.05), but the difference was not obvious between 15 and 18 °C and between 21 and 24 °C (*p* > 0.05) (Figure 3i,j).

In order to further explore the reasons for the decrease in MO_2_, the HGB of Amur grayling was measured (Table 3). The HGB increased significantly at 12–18 °C but decreased considerably in the 18–24 °C group (*p* < 0.05).

### 3.3. Changes in Transcriptome Profile Induced by Temperature Stress

A total of 37.73 Gb clean reads were generated in this project (Appendix A). They showed that 72403, 53508, 46471, 37509, 51687, and 50335 unigenes were mapped to Nr, eggNOG, Swiss-Prot, Pfam, Gene Ontology (GO), and KEGG databases, respectively (Appendix A). Principle component analysis (PCA) demonstrated marked discrepancy among different groups (Figure 4a). A total of 6251 DEGs (3263 up-regulation and 2988 down-regulation) and 80647 DEGs (33678 up-regulation and 46969 down-regulation) were found in C15 (15 °C) vs. Con (9 °C) and C21 (21 °C) vs. Con (9 °C), respectively (Figure 4b,c). According to GO analysis, unigenes in both 15 °C vs. 9 °C and 21 °C vs. 9 °C were assigned to the three GO domains (biological process, cellular component, and molecular function; Figure 4d,e), with cellular and metabolic processes constituting the dominant biological process categories. The main GO terms in cellular components were membrane and cell part. In the molecular function, binding and catalytic activity were the most enriched GO terms. KEGG enrichment analysis showed that all unigenes were assigned to 92 functional types, such as “Amino acid metabolism”, “Energy metabolism”, “Lipid metabolism”, and “Immune system” (Figure 4f–i).

### 3.4. GO Enrichment and KEGG Pathway Analysis for DEGs

DEGs between the 9 °C and 15 °C groups were enriched in the GO term biological process, cellular component, and molecular function categories, and the top 20 GO terms are shown in Figure 5a. In the liver of *Thymallus grubii*, the most representative DEGs annotated to biological process were protein localization to endoplasmic reticulum; the most dominant GO term mapped to cellular component was membrane; in molecular function, the most primary GO term was nutrient reservoir activity. GO enrichment analysis (Figure 5b) revealed that DEGs (21 °C vs. 9 °C) were largely enriched into regulation of biological process, cellular components, and molecular function. Among the enrichment, biological process was mainly the RNA metabolic process and RNA processing; the main category of cellular component and molecular function was mitochondrial protein-containing complex and RNA binding, respectively.

The KEGG pathway analysis showed that protein processing in the endoplasmic reticulum, protein export, and steroid biosynthesis were enriched mainly in the 15 °C vs. 9 °C comparison (Figure 5c). Oxidative phosphorylation, non-alcoholic liver disease, and spliceosomes were enriched mainly in the 21 °C vs. 9 °C comparison (Figure 5d). The fatty-acid degradation pathway was predominant in 21 °C vs. 9 °C. In addition, the pathways linked to protein processing or amino acid metabolism, such as glycine, serine, and threonine metabolism, N-glycan biosynthesis, and valine, leucine, and isoleucine degradation were included in both pairwise comparisons. KEGG enrichment analysis also identified glycolysis/gluconeogenesis, citrate cycle (TCA cycle), and apoptosis to be significantly enriched. Specifically, glycolysis/gluconeogenesis (*hk*)-, citrate cycle (TCA cycle) (*sdh, idh, aco*)-, fatty-acid biosynthesis (*fasn*)-, and fatty-acid degradation (*hadh, hadha*)-related genes were identified (Appendix A). Combined transcriptomic and physiological evidence indicates that warming affected the oxygen transport and energy metabolism in the liver and muscle tissue (Figure 6).

### 3.5. Validation of RNA-Seq Profiles

To verify the credibility of the transcriptome sequencing results, nine DEGs were randomly chosen (Appendix A). It has been shown that the trends of the changes in these candidate genes are in agreement with the results provided by the transcriptome project.

## 4. Discussion

Temperature change is a critical challenge for fish survival. It can influence the metabolism mode, including overall metabolic rate and oxygen consumption, and reduce nutrient digestion and absorption efficiency [31,32].

Oxygen consumption directly affects efficient energy usage and reflects overall physiological function under stress conditions [33]. It was found that MO_2_ in rainbow trout (*Oncorhynchus mykiss*) increased with rising environmental temperature [34]. Between 9 and 15 °C, MO_2_ in *Thymallus grubii* rose significantly with temperature, reflecting the exponential increase in metabolic demand for growth and reproduction [35]. The inflection point at 15 and 18 °C—where MO_2_ reached its maximum—thus marks the upper limit of the species’ optimal metabolic window. Further warming to 21 °C led to a decline in MO_2_, accompanied by a parallel drop in liver glycogen and muscle creatine phosphate content. This suggests that depleted energy reserves constrain metabolic scope.

In addition, the most pronounced changes in energy metabolism enzyme activities occurred between 12 and 15 °C, indicating that this range is vital for the acclimation process of *Thymallus grubii* to acute warming. In addition, genes associated with the hemoglobin complex (GO: 0005833), oxidoreductase activity (GO:0016491), and oxygen carrier activity (GO:0005344) were up-regulated after exposure at 15 °C, which is in accordance with the results of landscape transcriptomic analysis in wild brook trout (*Salvelinus fontinalis*) under heat stress [36]. Because warmer water holds less dissolved oxygen and requires higher respiration rates to maintain activity levels, genes (*hbad*, *hbb1*, *hba*, *hba4*, *hbb*, *hbb2*, and *kdm6a*) involved in oxygen transport and the content of HGB increased at 15 °C, which is the critical metabolic window of Amur grayling. Other studies have reported increased HGB and oxygen carrier activity in salmonids acclimated to high temperatures [37]. Nevertheless, there are also studies that found HGB content can decrease following shorter periods of acute temperature stress more similar to the heat stress in this study [38,39]. It was observed that only the hemoglobin complex (GO: 0005833) enriched in the 21 °C and HGB decreased above 21 °C, which reduced respiratory efficiency for Amur grayling. The result of this study is consistent with recent wild environment-based thermal tolerances [36]. It is evident that experimental methods which incorporated flow rate into temperature provide an effective experimental model for field testing. We coupled a Loligo swim-tunnel respirometer with an automatic temperature-control unit so that water temperature could be raised at 2 °C/h while flow velocity was held constant at 34 cm/s (Figure 1). This simultaneous control of both thermal and hydraulic variables mimicked the acute warming and steady current velocity conditions of the Amur grayling experience in mountain rivers. Because MO_2_, enzyme activities, and gene expression were recorded in real time, the setup provides a field-realistic, high-throughput model that could be deployed directly in river-side mobile laboratories for heat sensitivity testing of other stenothermal species.

PK and HK are key regulators of energy balance [40]. In *Thymallus grubii*, hepatic HK and PK activities peaked at 15 °C and then declined sharply above 18 °C, mirroring the temperature-dependent switch from aerobic to anaerobic metabolism. Reduced glucose availability under warming further suppressed HK activity, whereas PK remained elevated between 12 and 15 °C to accelerate glycolysis and replenish ATP for stress responses. These kinetics parallel observations in blue tilapia under acute warming [41]. Additionally, although HK did not reach statistical significance, its activity rose modestly from 9 to 15 °C and declined thereafter, mirroring the significant PK peak observed at the same temperatures. This parallel behavior supports the notion that HK and PK remained elevated between 12 and 15 °C to accelerate glycolysis and replenish ATP for stress responses and down-regulated once the critical metabolic window (>15 °C) was exceeded. LDH content and activity increased progressively from 12 to 18 °C, consistent with a compensatory shift towards anaerobic glycolysis as hemoglobin–oxygen affinity decreased [42]. Conversely, SDH activity declined continuously beyond 18 °C, confirming suppressed aerobic respiration. Blood glucose decreased steadily from 12 to 24 °C, indicating rapid utilization to fuel stress-induced metabolic demands. Liver and muscle glycogen peaked at 12–15 °C and then declined, demonstrating that acute warming first promotes glycogen deposition and subsequently accelerates glycogenolysis to maintain glucose supply. Comparable glycogen depletion has been reported in Chinese bream exposed to rising temperatures [43]. Between 9 and 15 °C, elevated aerobic efficiency provided ample energy, promoting the conversion of blood glucose into glycogen in the liver and muscles, with concurrent up-regulation of glycogen synthase, leading to glycogen accumulation and a simultaneous fall in blood glucose. When temperatures exceeded the critical thermal threshold of Amur grayling (15 °C), aerobic metabolism was suppressed (MO_2_ inflection), energy availability became limited, and rapid glycogenolysis mobilized hepatic and muscular glycogen to glucose, which was then channeled through enhanced glycolysis to meet urgent energy demands, perpetuating the decline in blood glucose. Transcriptomic analysis revealed that DEGs involved in the TCA cycle were down-regulated at 15 °C, suggesting that warming inhibits mitochondrial oxidative metabolism and redirects carbon flux towards glycolysis. Collectively, these data establish 15 °C as the upper boundary of the optimal thermal range for *Thymallus grubii* and highlight the pivotal role of rapid energy redistribution in coping with warming.

In order to physiologically compensate for environmental stressors, fish need to utilize energy substrates [44]. In the present study, transcriptomic analysis revealed that fatty-acid degradation, fatty-acid metabolism, and fatty-acid biosynthesis pathways were significantly enriched in the 15 °C and 21 °C groups. Fish can therefore fine-tune lipid metabolism to maintain homeostasis under warming. FASN, a multi-functional enzyme, catalyzes fatty-acid biosynthesis, whereas HADH and HADHA are key genes involved in fatty-acid degradation [45]. In this study, the expression level of *fasn* increased in the 15 °C group, whereas those of *hadh* and *hadha* decreased. As temperature increased, the opposite trend was observed in the 21 °C group. The results suggest that 21 °C inhibited the lipid synthesis pathway and facilitated the fatty-acid degradation, presumably to meet the elevated energy demand by regulating lipid metabolism in response to high temperature stress. Above all, fatty-acid metabolism in fish may play an important role in adaptation to environmental temperature stress [46]. Meanwhile, *Thymallus grubii* significantly regulated amino acid, lipid, and glucose metabolism pathways to defend against physiological perturbations caused by acute warming.

In fish, temperature affects protein synthesis, and a clear relationship exists between the optimal temperature range for survival and modifications in protein metabolism [47]. In the present study, transcriptomic analyses revealed that the expression levels of *nef*, *calr*, and *erp27* were down-regulated within the endoplasmic reticulum protein-processing pathway. Similar reductions were observed for genes involved in protein export, including those encoding the protein transport proteins *sec61α* and *sec61β*, the signal peptidase complex subunits *spcs1* and *spcs2*, and the signal recognition particle subunits *srp14* and *srp68*. Comparable patterns have been reported across a range of fish species [48]. Compared with the 9 °C group, the 15 °C group exhibited markedly reduced hepatic concentrations of alanine, aspartate, glutamate, glycine, serine, and threonine, results that align with observations in grass carp (*Ctenopharyngodon idellus*) [49]. Exposure to a high temperature of 15 °C decreased amino acid synthesis efficiency, thereby inhibiting protein synthesis in *Thymallus grubii*. Because the TCA cycle is fueled by amino acids such as aspartate, alanine, glutamate, serine, glycine, and threonine, the reduced availability of these substrates is a major reason for the observed down-regulation of the TCA cycle in the 15 °C group. Previous studies have shown that elevated temperatures increase hepatic ammonia concentrations as a consequence of accelerated protein and amino acid catabolism, a process that can damage liver tissue and compromise fish health [50,51]. Glutamate is the principal form of deamination in fish. In contrast to the decline in other amino acids, glutamate levels were elevated in the present study. These findings diverge from those reported for steelhead trout (*Oncorhynchus mykiss*) and Atlantic salmon (*Salmo salar*) [52,53], possibly owing to species-specific thermal stress thresholds.

In natural environments, grayling exhibit stage-specific temperature preferences throughout their life history. For European grayling (*Thymallus thymallus*), the preferred summer temperature ranges from 12.0 to 15.7 °C [19], whereas Arctic grayling (*Thymallus arcticus*) thrive at 6–12 °C during the same season [54]. In the present study, the optimal physiological thermal limit for Amur grayling did not exceed 15 °C, a threshold that closely matches the upper thermal limits documented for other Thymallus species in nature. These findings demonstrate a tight coupling between physiological requirements and temperature across the examined life-history stages, a conclusion further corroborated by our laboratory investigations into the underlying physiological response mechanisms of grayling.

As water temperature increases, cold-water populations retreat to higher, cooler thermal refuges; however, such refuges are limited in extent and therefore reduce the amount of suitable habitat [55]. Studies have shown that warming within or beyond the temperature preference and tolerance ranges of bull trout (*Salvelinus confluentus*) and brook trout (*Salvelinus fontinalis*) results in the loss of thermally suitable natal habitat [56,57]. Consequently, it is hypothesized that Amur grayling will seek cooler habitats once the water temperature exceeds 15 °C, provided that such refuges are accessible. In addition, elevated temperatures can shift the timing of grayling spawning either earlier or later, disrupting synchrony with food resources and thereby reducing the survival of embryos, larvae, and fry, which ultimately contributes to population decline [17,58]. Hauer et al. [21] documented that grayling spawning occurs within 7.8–11.5 °C. It can therefore be reasonably deduced that reproduction may be impaired when the water temperature exceeds 15 °C, thereby accelerating the risk of population extinction.

## 5. Conclusions

Our study provided comprehensive overviews for the detailed effect of change in warming on energy metabolism mode in *Thymallus grubii* based on transcriptome technology. Our results indicate that warming contributed to changes in transcriptome profiles, as well as a series of metabolism imbalance responses mediated by enzymatic reaction, which may be appropriate for assessing the health status of Amur grayling in natural environments to deal with temperature changes. Meanwhile, *Thymallus grubii* may be able to maintain physiological homeostasis under high thermal challenge through activation of the gluconeogenesis process and regulation of lipid/amino acid metabolic pathways. Our study provided basic information for exploring the molecular mechanism of warming on cold water fish. Our results suggest that acute exposure of adult Amur grayling to temperatures above approximately 15 °C during active swimming induces substantial physiological stress, which supports existing concerns about summer warming in their habitats. However, chronic exposure (days to weeks) and early life stages (eggs, larvae, and juveniles) were not examined, and longer-term effects on growth, reproduction, and lifetime fitness remain unknown. We therefore recommend that 15 °C be used only as an acute warming threshold for river management until chronic and ontogenetic thermal limits are established through prolonged warming experiments and field validation. Further work is vital to study how to increase Amur grayling population resilience via restoring natural discharge regimes and limiting future temperature increases [23,59].

## Figures and Tables

**Figure 1 biology-14-01718-f001:**
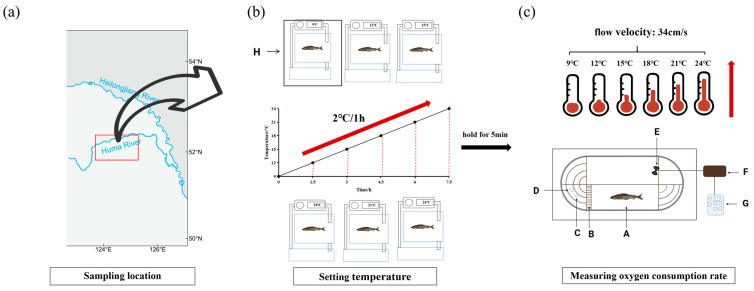
Sampling flow chart of Amur grayling during warming. (**a**) Sampling location, (**b**) thermal equilibration, (**c**) schematic diagram of the flume-type swimming respirometer. (A) Fish swimming area, (B) rectifier, (C) dissolved oxygen (DO) probe, (D) temperature sensor, (E) propeller, (F) propeller motor, (G) frequency changer, (H) temperature-controlled recirculating water tank.

**Figure 2 biology-14-01718-f002:**
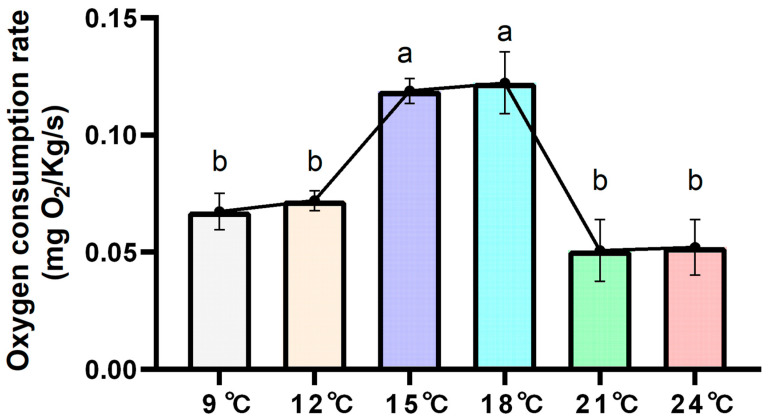
The relationship between oxygen consumption rate (MO_2_) and water temperature (T) (n = 8). Different lowercase letters represent significant differences (*p* < 0.05) between groups (mean ± SD).

**Figure 3 biology-14-01718-f003:**
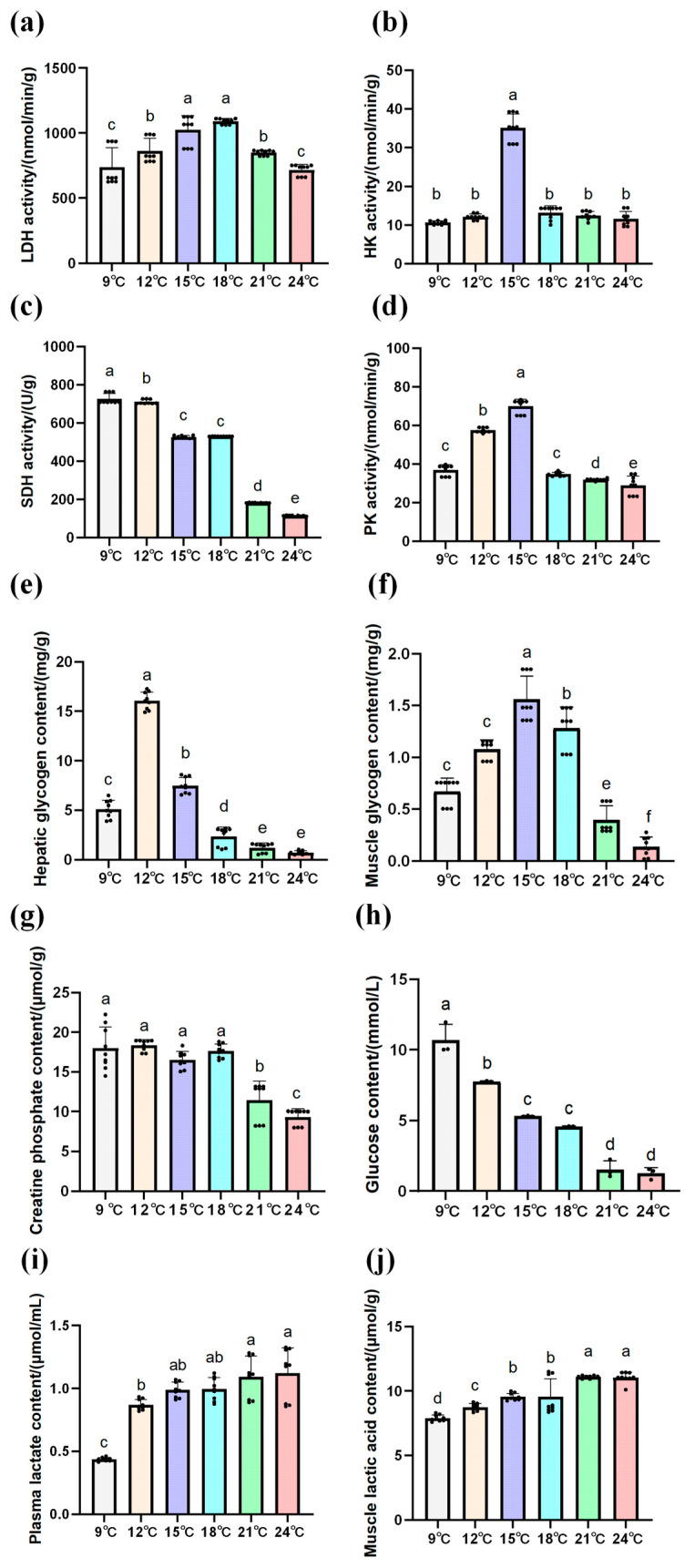
Effects of carbohydrate metabolism on enzyme activities and contents of biochemical substances in liver tissues under warming (n = 3). (**a**) LDH activity. (**b**) HK activity. (**c**) SDH activity. (**d**) PK activity. (**e**) The content of liver glycogen. (**f**) The content of muscle glycogen. (**g**) The content of creatine phosphate. (**h**) The content of blood glucose. (**i**) The content of plasma lactic acid. (**j**) The content of muscle lactic acid. Different lowercase letters represent significant differences (*p* < 0.05) among different temperature groups (mean ± SD). A one-way ANOVA test was used to identify the statistical significance.

**Figure 4 biology-14-01718-f004:**
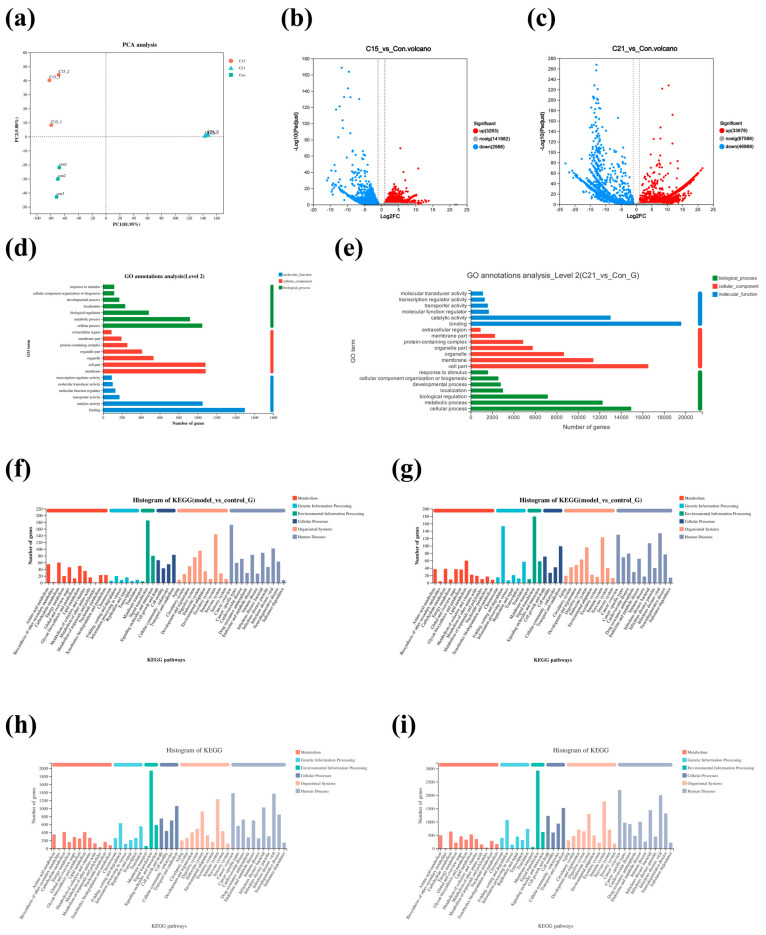
Non-reference genome-based transcriptome analysis for liver under warming (n = 3). (**a**) Principal component analysis (PCA) scatter diagram. The 9, 15, and 21 °C groups were represented by green squares, orange circles and blue triangles, respectively. (**b**) Volcano map of the distribution trends of differentially expressed genes (DEGs) between 15 and 9 °C. Each dot represents one gene. Blue and red dots represent DEGs. Gray dots represent non-differentially expressed genes. (**c**) Volcano map of the distribution trends of differentially expressed genes (DEGs) between 21 and 9 °C. (**d**) Bar chart showing the significantly enriched GO terms in 15 °C vs. 9 °C. (**e**) Bar chart showing the significantly enriched GO terms in 21 °C vs. 9 °C. The results for 15 vs. 9 °C and 21 °C vs. 9 °C are divided into three main categories: cellular components, biological process, and molecular. The X-axis at the bottom indicates the number of unigenes. The Y-axis indicates the subcategories. (**f**,**g**) Bar chart of enriched KEGG pathways of up-regulated and down-regulated genes of 15 °C vs. 9 °C. (**h**,**i**) Bar chart of enriched KEGG pathways of up-regulated and down-regulated genes of 21 °C vs. 9 °C. Con: control group (9 °C); C15: 15 °C group; C21: 21 °C group.

**Figure 5 biology-14-01718-f005:**
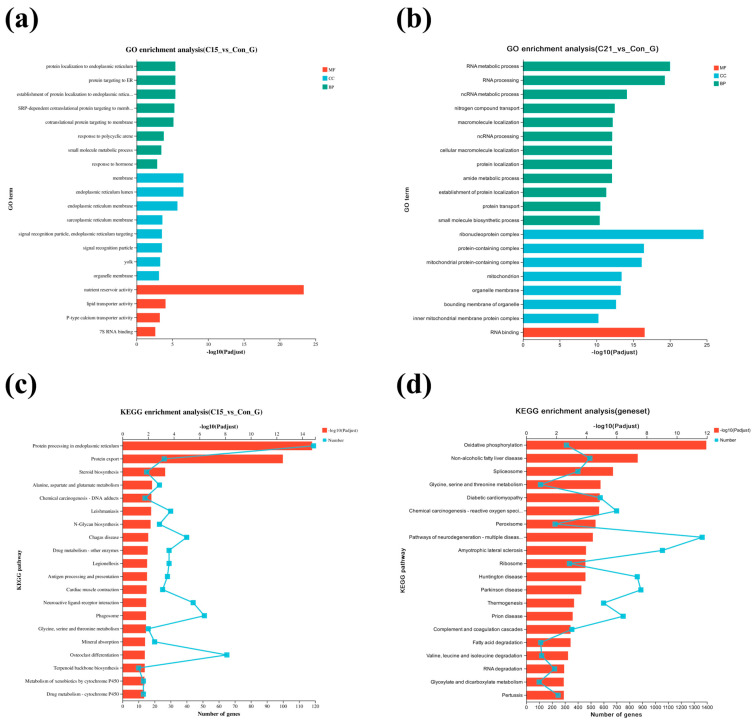
Enrichment analysis of gene GO term and KEGG pathway for DEGs under warming. (**a**,**b**) Bar chart of the top 20 significantly enriched DEGs of the two comparison groups in Amur grayling. The two comparison groups include 15 °C vs. 9 °C and 21 °C vs. 9 °C. The different colors in the GO chart represent the enriched categories with −log10(Padjust) as the standard. (**c**,**d**) KEGG enrichment of the two DEG comparison groups in Amur grayling. The two comparison groups include 15 °C vs. 9 °C and 21 °C vs. 9 °C. The red bar in the KEGG chart represents the enriched categories with -log10(Padjust) as the standard. The polylines represent the number of genes enriched into different categories. Con: control group (9 °C); C15: 15 °C group; C21: 21 °C group. KEGG pathway: © Kanehisa Laboratories; reproduced with permission. Individual pathway IDs are listed in Appendix A. establishment of protein localization to endoplasmic reticu…: establishment of protein localization to endoplasmic reticulum; SRP-dependent cotranslational protein targeting to memb…: SRP-dependent cotranslational protein targeting to membrane; Chemical carcinogenesis-reactive oxygen speci…: Chemical carcinogenesis-reactive oxygen species; Pathways of neurodegeneration-multiple diseas…: Pathways of neurodegeneration-multiple diseases.

**Figure 6 biology-14-01718-f006:**
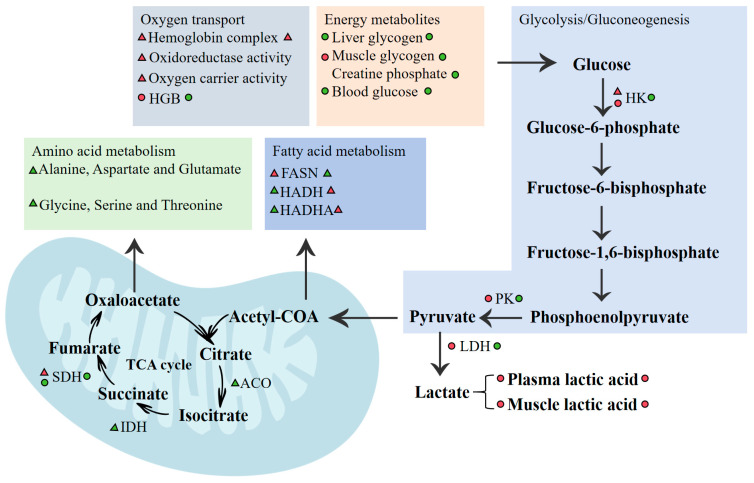
Schematic overview of the alteration in physiological parameters and transcriptome in the 15 and 21 °C groups of Amur grayling. Green and red represent up-regulation and down-regulation, respectively. Triangles represent DEGs. Circles represent physiological parameters. The shape on the left and right represent 15 and 21 °C, respectively. HGB: Hemoglobin concentrations; FASN: Fatty-acid synthase, animal type; HADH: 3-hydroxyacyl-CoA dehydrogenase; HADHA: Enoyl-CoA hydratase/long-chain 3-hydroxyacyl-CoA dehydrogenase; HK: Hexokinase; PK: Pyruvate kinase; LDH: Lactate dehydrogenase; ACO: Aconitate hydratase; IDH: Isocitrate dehydrogenase; SDH: Succinate dehydrogenase.

**Table 1 biology-14-01718-t001:** The size of target fish for the experiment.

Water Temperature	9 °C	12 °C	15 °C	18 °C	21 °C	24 °C
age/year	3^+^	3^+^	3^+^	3^+^	3^+^	3^+^
sample size	8	8	8	8	8	8
wet weight/g	50.43 ± 5.21 ^a^	55.84 ± 7.98 ^a^	54.39 ± 7.41 ^a^	54.67 ± 7.85 ^a^	53.93 ± 3.30 ^a^	54.92 ± 4.96 ^a^
body length/cm	16.98 ± 0.47 ^a^	17.00 ± 0.47 ^a^	16.91 ± 0.30 ^a^	17.03 ± 0.44 ^a^	17.16 ± 0.38 ^a^	17.13 ± 0.30 ^a^

Note: Different lowercase letters represent significant differences (*p* < 0.05) between groups. Data are presented as mean ± SD (n = 8). 3^+^: 3–4 years old.

**Table 2 biology-14-01718-t002:** Temperature coefficient Q_10_ values of respiration of Amur grayling at different temperatures.

Temperature/°C	Temperature Coefficient Q_10_
9–12	1.25
12–15	5.30
15–18	1.10
18–21	0.05
21–24	1.09

**Table 3 biology-14-01718-t003:** Effects of warming on hemoglobin concentrations (HGB) of Amur grayling.

Water Temperature	9 °C	12 °C	15 °C	18 °C	21 °C	24 °C
HGB (g/L)	96.07 ± 6.84 ^c^	102.62 ± 10.43 ^c^	117 ± 9.12 ^b^	127.78 ± 3.49 ^a^	37.9 ± 6.14 ^d^	42.29 ± 9.52 ^d^

Note: Different lowercase letters represent significant differences (*p* < 0.05) between groups. Data are presented as mean ± SD (n = 3).

## Data Availability

The datasets presented in this study can be found in online repositories. The names of the repository/repositories and accession number(s) can be found below: https://www.ncbi.nlm.nih.gov/bioproject/PRJNA1369758, accessed on 27 November 2025.

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
