# Peer review of "Integrated Oxygen Consumption Rate, Energy Metabolism, and Transcriptome Analysis Reveal the Heat Sensitivity of Wild Amur Grayling (Thymallus grubii) Under Acute Warming"

_biology, 2025, doi:10.3390/biology14121718_

Round 1

Reviewer 1 Report

Comments and Suggestions for Authors

I am confident that this manuscript showed an interesting report with the presence of warming on the Oxygen Consumption Rate and Energy Metabolism in Wild Amur Grayling.

One major suggestion is that gill, liver, kidney and intestine histology and ultrastructure should be included. It is very important to explain the effect of warming on the physiological activity in Wild Amur Grayling.

For example,

Extreme warm acclimation temperature alters oxygen consumption, micronucleus formation in erythrocytes, and gill morphology of rohu (Labeo rohita) fingerlings

Alterations of oxygen consumption and gills morphology of Nile tilapia acclimatized to extreme warm ambient temperature

Effects of temperature on the oxygen consumption rate and gill fine structure of hybrid grouper, Epinephelus fuscoguttatus ♀ × E. Lanceolatus

I have several questions and unclear messages for reader, as listed below.

Abstract and summary

Unclear keywords such as breeding temperature (Line 37) hepatic glycogen content (Line 41) quantitative guidelines for conservation (Line 47) are found. More explanations about them terms should be added.

Introduction

It is not satisfactory for me, and I am fine confusing, especially several text and logical flow refer to the ordered arrangement of data.  They are mixed and I suggest revising the following.

  1. Water temperature - the catastrophic mortality- poikilothermic animals - fishes
  2. Impact of water temperature - dissolved oxygen, genetics, physiology, behavior, and environmental adaptability of fish (Line 70)
  • Wild Amur Grayling (Thymallus grubii) – report about the biology, gap analysis and objective!

Additionally, important keywords such as Heat Sensitivity and Critical Thermal Preference were not found, and I am sure that it is not good if these words are not mentioned in the introduction.

Methods

The reason what the authors used it is is not clear.

How can the experiment be replicated?

Please specify life stage of fish in the experiment.

Please summarize the data in table showing the parameters that the authors used.

What is the wet weight in Table 1? Total weight?

Results

The results are clear and suitable for publication, but some data are found.

Please re-write section 3.2 and I suggest that the authors should be separated the parameters for physiological index. They are mixed and difficult to read.  

Fragmented sentences are found such as Line 281-282. Please re-check throughout the manuscript.

What is the meaning “such as “Amino acid metabolism”, “Energy metabolism”, “Lipid metabolism”, and “Immune system” (Line 285-286)

Figure 6 is the critical message, and I am not fully convinced that all parameters are included of this project. No data when the author showed in the mitochondrial pathway. Are you sure that this is an actual result from the study?

Enzyme Activity of Glucose Metabolism is first show in the results, not refer to the introduction

The discussions points are mixed into the results such as Line 308-309. Only results are present.

More explanation in Line 322-323. What is the evidence of integrated analysis between A and B!.

The glucose in 3h, it is dramatically decrease from 9 C to 24 C and all sub-figures showed a trend toward a critical point during 15-18 C. The authors should focus these periods for discussion.  

The glycogen in liver and muscle is opposite. Is it possible to mention the site of glycogen od the target fish?

I am not sure that the number for used in each experiment and the authors mentioned “eight” in the text and Figure 3 “tree samples” Could the authors clarify this discrepancy?

The expression of PK and HK (Figure 3b and 3d) is similarly found and should be added the point in the discussion.

Discussion

What is the point of Warming on the Oxygen Consumption - Heat Sensitivity and Critical Thermal Preference How?

Line 362 - in-vitro – italic

Line 364-365

The author mentioned “ In addition, the most pronounced changes in energy metabolism enzyme activities occurred between 12 and 15℃, indicating that this range was vital for the acclimation process of Thymallus grubii to warming ….. What are the evidence?

Line 371 – please specific gene type

Line 378-379 – How?

Reviewer 2 Report

Comments and Suggestions for Authors

The topic fits the scope of a biology / fish physiology journal, and the combination of in-swim-flume respirometry with biochemical and transcriptomic data on a wild, conservation-relevant salmonid is potentially valuable. However, in its current form the manuscript has several substantial problems:

  1. Inadequate and sometimes confusing description of the experimental design, especially the definition of the transcriptomic comparison groups and exposure regime.
  2. Questionable treatment and interpretation of the MO₂–temperature relationship, including the use of a very poor exponential fit (R² = 0.102) that appears to be mathematically inconsistent with the described pattern.
  3. Over-interpretation of results in terms of “critical thermal limit” and “preference” without using standard CTmax/CTmin or behavioral preference approaches.
  4. Some discussion paragraphs read more like a generalized review and at times assert results that were not actually measured here (for example, mention of muscle ATP content).
  5. Several key pieces of information are missing or not clearly presented (data deposition, details of respirometry protocol, exact definition of groups Con, C15, C21).
  6. Language and presentation need thorough editing for clarity and scientific precision

On the positive side, the core idea and the data set look salvageable. With substantial revision and clarification, this could become a solid contribution, but it is not ready for acceptance now.

A negative exponent in this form implies a monotonic decline of MO₂ with temperature, which directly conflicts with the stated pattern “initially elevated and then decreased with increasing water temperature”.

A standard approach in this field is to provide Q₁₀ values for MO₂ between temperature intervals and, where possible, an estimate of routine vs maximum metabolic rate to discuss aerobic scope. Here, the interpretation of “critical metabolic window” and “upper limit” would be much stronger if supported by Q₁₀ calculations and a more explicit treatment of scope.

A decline in MO₂ above 15–18 °C is interpreted as “metabolic suppression” and linked to both depleted energy reserves and possible systemic hypoxemia. However, with a very short acute exposure, changes in MO₂ could also reflect behavioral changes in swimming effort within the flume or incipient fatigue.

Fish are moved from 9 °C to target temperatures at 2 °C h⁻¹, held 5 minutes at the new temperature, then transferred to the swim chamber and measured for 40 minutes. This is a short acute exposure, not an acclimation. Yet the paper repeatedly refers to “critical thermal limit ranges”, “critical thermal preference limit” and “optimal thermal window for growth and reproduction”.

The text says: “A total of 6251 DEGs was found in C15 (9 °C) vs Con (15 °C) and 80647 DEGs in C21 (9 °C) vs Con (15 °C).” This is logically inconsistent. C15 and C21 should correspond to 15 and 21 °C, not 9 °C. Figures use “Con, C15, C21” but the labels and legend do not clearly define which temperature each represents. Precisely define Con, C15, C21 once in Methods and in each figure legend. Make sure the transcriptomic comparisons are clearly 9 vs 15 °C and 9 vs 21 °C (if that is indeed the case) or 15 vs 9, 21 vs 15, etc.

For the swim-flume respirometry, important features are missing:

  • Was intermittent-flow or continuous-flow respirometry used

  • Duration of each measurement period and flush period

  • Minimum DO level allowed before flushing

  • Whether background respiration was measured explicitly and subtracted

  • Whether fish were fasted for a standard period beyond the initial 3 days

  • Exact MO₂ units used in figures and text

The sentence “MO₂ in the system was less than 1% of Amur grayling MO₂ when no fish were in the respirometer, therefore, its error impact could be ignored” is very brief and does not explain how this was evaluated.

The text refers to “muscle ATP content” in the discussion, but ATP was not measured. Only creatine phosphate was quantified. Several statements about “systemic hypoxemia”, “cellular damage” and “mitochondrial suppression” are made without direct supporting data from this study.

The MO₂ results section says that 15 °C and 21 °C were selected as “test groups of transcriptome and untargeted metabolomics analysis”, but the manuscript only presents transcriptomic results. Either provide at least a summary of the metabolomics results or remove references to untargeted metabolomics from the main text and abstract. For RNA-seq studies, most journals require deposition of raw reads in a public repository (NCBI SRA, EBI, etc) with accession numbers in the manuscript. The current statement “available from the corresponding author upon reasonable request” is not sufficient for this type of work.

The conclusion that “cultivated and natural temperatures should be controlled under 15 °C” is very strong given that the fish experienced high temperatures for less than an hour and only as adults under sustained swimming. Long-term fitness, growth and reproduction may require longer exposures and life-stage specific tests. Reframe this as “our results suggest that acute exposure of adult Amur grayling to temperatures above approximately 15 °C during active swimming induces substantial physiological stress, which supports existing concerns about summer warming in their habitats.” Emphasize that further work on chronic exposures and early life stages is needed before firm temperature thresholds for management can be defined.

Throughout, there are numerous grammatical errors and awkward phrases. Examples include “were affected by increased environment temperature that was caused by global warming”, “while the greatest sensitivity was exhibited between 12–15 °C”, “would affect the survival, reproduction, growth performance, and physiological parameters of cold water fish”.

Use “water temperature” or “temperature” consistently, rather than alternating with “breeding temperature” for acute experimental temperatures.

Always include units for MO₂ in figures and text. Hemoglobin is reported as 37.9 ± 6.14 g L⁻¹ at 21 °C and 42.29 ± 9.52 g L⁻¹ at 24 °C, compared to ~100–128 g L⁻¹ at lower temperatures. These very low values at high temperature are striking and deserve explicit comment in the results and discussion, including whether any measurement artefacts are possible.

A few references are drawn from human oncology or other distant fields to support statements on lipid metabolism (for example, melanoma, AML papers). These are not ideal for a fish physiology paper when fish-specific literature exists.

The study has a promising design and uses a valuable combination of approaches on an ecologically important fish. However, the current manuscript has significant issues in experimental description, data analysis, and interpretation.

Comments on the Quality of English Language

A thorough edit by a fluent English speaker or professional editing service is strongly needed.

Reviewer 3 Report

Comments and Suggestions for Authors

Thanks for the editor's invitation. I have conducted a comprehensive review of the manuscript titled "Effect of warming on the oxygen consumption rate and energy metabolism in wild Amur grayling (Thymallus grubii): Revealing the heat sensitivity and critical thermal preference limit". This research explored the physiological and molecular responses of a rare cold-water fish to increasing temperature and pointing to a critical comfort zone between 12℃ and 15℃. These findings provided valuable guidelines for the conservation of the wild Thymallus grubii population under the trend of global warming.

Major concern:

  1. While the study is well-structured, the Introduction and Discussion sections could be significantly shortened and focused to improve readability and impact. I recommend that the authors streamline these sections. Once these issues are sufficiently addressed, the manuscript has the potential to be accepted for publication.
  2. Furthermore, although global warming is an irreversible trend, the temperature gradient established in this study remains excessively wide, potentially leading to overestimation of the true ecological effects.
  3. Given the inverse relationship between water temperature and dissolved oxygen concentration, it is crucial to specify whether dissolved oxygen was monitored throughout the experiment in different treatment and the actual values should be tabulated in the results.

Line 66-77:It is recommended to remove this paragraph, given that water temperature constitutes the core experimental variable, whereas the effects of water flow on fish could be sufficiently addressed with a concise overview.

Line 136-138: Please indicate the duration of artificial rearing/acclimation that the experimental fish underwent prior to the experiment.

Line 165-167: Could the authors justify the selected warming rate of 2°C per hour? This rate appears substantially faster than the gradual temperature shifts typically observed in natural aquatic ecosystems.

Line 167: Should be Figure 1b here.

Line 211-218: I recommend consolidating these sections and providing a concise description of the methods, rather than directly stating "Detailed procedure could be referred to Appendix...".

Line 234: I do not get why the authors fitted an exponential function, as the low R-squared value may indicate insufficient reliability.

Round 2

Reviewer 1 Report

Comments and Suggestions for Authors

All comments are nice and it is ready to go for publications. I suggested that the histological figures of selected organ should be added as the supplymentary data. 

Best wishes,

Reviewer 2 Report

Comments and Suggestions for Authors

After the corrections, the manuscript can be accepted for publication

Comments on the Quality of English Language

A thorough edit by a fluent English speaker or professional editing service is strongly needed.

Reviewer 3 Report

Comments and Suggestions for Authors

I think the manuscript has been significantly improved and now meets the standards for publication. The authors have diligently addressed all major concerns raised during the review, particularly by strengthening the methodological description and refining the discussion of the results. The paper presents a valuable contribution to the field, and I recommend its acceptance in its current form. Thanks to the authors on their good work.